# Dynamic Facial Emotional Expressions in Self-Presentation Predicted Self-Esteem

**DOI:** 10.3390/bs15050709

**Published:** 2025-05-21

**Authors:** Xinlei Zang, Juan Yang

**Affiliations:** 1Faculty of Psychology, Southwest University, No. 2 Tiansheng Road, Beibei, Chongqing 400715, China; terryzang@email.swu.edu.cn; 2Key Laboratory of Cognition and Personality, Southwest University, Chongqing 400715, China

**Keywords:** self-esteem, emotional expression, dynamic emotion, machine learning

## Abstract

There is a close relationship between self-esteem and emotions. However, most studies have relied on self-report measures, which primarily capture retrospective and generalized emotional tendencies, rather than spontaneous, momentary emotional expressions in real-time social interactions. Given that self-esteem also shapes how individuals regulate and express emotions in social contexts, it is crucial to examine whether and how self-esteem manifests in dynamic emotional expressions during self-presentation. In this study, we recorded the performances of 211 participants during a public self-presentation task using a digital video camera and measured their self-esteem scores with the Rosenberg Self-Esteem Scale. Facial Action Units (AUs) scores were extracted from each video frame using OpenFace, and four basic emotions—happiness, sadness, disgust, and fear—were quantified based on the basic emotion theory. Time-series analysis was then employed to capture the multidimensional dynamic features of these emotions. Finally, we applied machine learning and explainable AI to identify which dynamic emotional features were closely associated with self-esteem. The results indicate that all four basic emotions are closely associated with self-esteem. Therefore, this study introduces a new perspective on self-esteem assessment, highlighting the potential of nonverbal behavioral indicators as alternatives to traditional self-report measures.

## 1. Introduction

Self-esteem is a core concept in psychology with profound implications for individuals ([31]). It is defined as a person’s subjective evaluation of self-worth and the emotional experiences associated with it ([11]). According to this definition, high self-esteem is characterized by more positive emotions and fewer negative emotions. Research has demonstrated a strong association between self-esteem and self-relevant emotions ([6]). In their study, [6] ([6]) measured individuals’ emotions using the Positive and Negative Affect Schedule (PANAS) and assessed self-esteem using the Rosenberg Self-Esteem Scale. Their findings revealed significant correlations between self-esteem and all items in the PANAS. The emotions measured by PANAS reflect long-term and stable emotions (affective states) ([54]). Beyond such stable emotional tendencies, emotions can also manifest as transient affective states ([53]) and momentary emotional expressions ([16]). However, due to the limitations of traditional psychological measurement tools (e.g., self-report questionnaires), it remains challenging to capture real-time momentary emotional expressions. Consequently, the relationship between self-esteem and dynamic, transient emotional expressions is still not well understood.

Within the framework of Basic Emotion Theory (BET), emotions are conceptualized as distinct, short-lived states involving physiological, subjective, and expressive components that enable adaptive responses ([12]; [13]). According to BET, emotional expressions are brief yet coherent behavioral patterns that co-vary with subjective experiences, providing insight into an individual’s current emotional state ([14]; [42]). Emotions can be conveyed through facial expressions, vocalizations, and bodily movements ([41]), with facial expressions serving as a crucial medium for emotional communication.

Understanding individuals’ emotional states through facial expressions is a central research topic in the field of affective computing ([35]; [52]). Experts in affective computing analyze micro-expressions to uncover individuals’ concealed emotions, with applications in lie detection and criminal investigations ([5]). Micro-expressions are brief (<500 ms), involuntary facial expressions that involve minimal facial muscle activation and low intensity ([57]). In contrast to micro-expressions, individuals can consciously control macro-expressions, which conform to social norms and typically last between 0.5 and 4 s, engaging multiple facial muscles ([57]). Both micro-expressions and macro-expressions can effectively represent the six basic emotions—happiness, sadness, fear, disgust, anger, and surprise—described in BET ([14]; [39]). These emotions can be reliably identified through specific Facial Action Units (AUs), as summarized in Table 1 ([27]). In recent years, psychological research has shifted its focus from static emotional expressions to dynamic emotional expressions ([22]; [1]). Therefore, rather than concentrating solely on long-term and static emotions (such as affect), greater attention should be given to the study of dynamic emotional expressions.

Existing research has established a close relationship between self-esteem and behavioral expressions, particularly in the case of happiness-related expressions such as smiling. [29] ([29]) presented standardized and spontaneous posture photographs of participants to complete strangers (absolute zero-acquaintance) and asked them to evaluate various personality traits. Their findings revealed that individuals with higher self-esteem appeared healthier, smiled more frequently, exhibited more dynamic body postures, and tended to place their arms behind their backs while standing. Similar findings were observed in a related study where participants’ self-introduction videos were presented to complete strangers ([20]). [20] ([20]) further identified intense smiling, friendly facial expressions, charming/flirtatious behavior, and alert facial expressions as effective cues for self-esteem. These studies collectively suggest a strong association between self-esteem and smiling. Within the framework of BET, the facial expression of happiness is primarily characterized by smiling, which corresponds to the activation of AU6 (cheek raiser) and AU12 (lip corner puller). Moreover, individuals with low self-esteem are more prone to displaying negative emotions, such as sadness ([33]).

While prior studies have examined the relationship between self-esteem and facial expressions, they have predominantly focused on static macro-expressions. To the best of our knowledge, no studies have yet explored the dynamic aspects of emotional expressions in relation to self-esteem. One reason for this gap is the reliance on manual facial expression coding in traditional self-esteem research. While manual coding methods offer strong interpretability, they fall short in capturing subtle dynamic facial expressions, such as micro-expressions lasting less than 500 ms. Since such fine-grained emotional dynamics exceed human perceptual capacity, investigating dynamic facial expressions has been particularly challenging. As a result, our understanding of self-esteem-related dynamic emotional expressions remains limited.

This study aims to introduce affective computing techniques to uncover the relationship between self-esteem and dynamic emotional expressions. First, we employ computer vision technology to capture individuals’ AUs and compute momentary emotional expressions based on the framework of BET. Next, we apply time-series analysis to extract the dynamic features of these transient emotions. We then construct a model linking self-esteem with dynamic emotional features using machine learning techniques. Finally, through the popular explainable AI technique known as SHapley Additive exPlanations (SHAP; [25]), we examine which basic emotion’s dynamic features are closely associated with self-esteem.

## 2. Methods

Our study builds upon previous research on the relationship between self-esteem and static emotional expressions ([6]), following a similar approach: measuring individual self-esteem and momentary emotions separately, then establishing their relationship. Consequently, the key technical challenge in this study is determining how to effectively measure momentary emotions. A second major challenge is constructing the relationship between self-esteem and momentary emotional expressions. Traditional static emotion measurements yield scalar values, allowing for straightforward correlation analyses with self-esteem. However, dynamic emotional features are considerably more complex, typically represented as multidimensional vectors (arrays). This complexity renders conventional analytical methods—such as regression or analysis of variance (ANOVA)—inapplicable. Therefore, novel computational approaches are required to explore the intricate relationship between self-esteem and dynamic emotional expressions.

Based on the definition of self-esteem and previous research findings ([6]; [11]), self-esteem is most closely associated with self-relevant emotions. Therefore, it is essential to elicit self-relevant emotional experiences and dynamic emotional fluctuations in participants. A crucial consideration in our study is that our goal is to measure dynamic emotional expressions rather than emotional experiences induced by external stimuli. This means that the emotional responses should be spontaneous and dynamic, rather than a direct reaction to a specific emotional stimulus. To achieve this, we adopted a self-presentation task, a well-established paradigm that aligns with our research objectives ([20]; [51]). First, emotions experienced during self-presentation are inherently self-relevant, fulfilling our requirement for self-esteem-related emotions. Second, the process of self-presentation is spontaneous and natural, ensuring the dynamic nature of emotional expression. Additionally, to further amplify the role of self-esteem, we required participants to engage in self-presentation in front of an audience. Prior research has shown that public self-presentation intensifies behavioral differences between individuals with high and low self-esteem ([4]), making it an ideal setting for our study.

### 2.1. Participants

A total of 211 undergraduate and graduate students from Chongqing, China, were recruited online to participate in the experiment. The sample included 33 male participants (*M*_age_ = 20.55, *SD* = 2.21) and 178 female participants (*M*_age_ = 20.20, *SD* = 1.80). All participants provided informed consent and received monetary compensation upon completing the study.

### 2.2. Experimental Procedure

The experimental procedure is illustrated in the flowchart in Figure 1A. Upon arriving at the laboratory, participants first completed a series of questionnaires, including the Rosenberg Self-Esteem Scale and measures of social evaluation threat. After completing these questionnaires, they proceeded to the public self-presentation task. Following the experimental task, participants filled out an additional set of questionnaires, including measures of social evaluation threat and the Positive and Negative Affect Schedule (PANAS). Upon completing the post-task questionnaires, participants were free to leave the laboratory.

In this study, the public self-presentation task required participants to introduce themselves for 40 s in front of two research staff members (one male, one female). There were no restrictions on the content of the introduction. Participants’ performances during the task were recorded in full using a Sony HDR-PJ675 digital video camera, and the recordings were saved in MP4 format (1280 × 720 resolution, 25 Hz frame rate). The laboratory setup and the spatial arrangement of participants are illustrated in Figure 1.

### 2.3. Experimental Measures

This study utilized the Rosenberg Self-Esteem Scale (RSES) to assess participants’ trait self-esteem ([38]) and the Positive and Negative Affect Schedule (PANAS) to measure their long-term, stable positive and negative emotions ([54]). Additionally, subjective social evaluation threat (SET) was assessed using a single-item self-report measure.

The RSES is a widely used scale designed to measure global self-esteem, focusing on individuals’ general self-perceptions rather than specific qualities or attributes. It consists of 10 items, rated on a 4-point Likert scale. In the Chinese version of the RSES, the eighth item is positively worded, meaning that four items require reverse scoring ([47]). The total self-esteem score is obtained by summing all item responses, with higher scores indicating higher self-esteem levels. The scale is widely recognized for its strong reliability and validity ([11]). In this study, Cronbach’s coefficient for the RSES was 0.90, indicating high internal consistency.

The PANAS is used to assess long-term, stable, and static emotions. The scale consists of 20 items, with 10 items measuring positive affect (PA) and 10 items measuring negative affect (NA), rated on a 5-point Likert scale. Higher scores on individual items indicate a stronger intensity of the corresponding PA or NA emotions.

The SET measure is adapted from the Trier Social Stress Test (TSST) paradigm, where participants are directly asked about their perceived social evaluation threat ([10]). This approach has been validated for use in Chinese populations ([50]). The SET measure consists of a single item, rated on a 7-point Likert scale, with higher scores indicating a greater perceived social evaluation threat.

### 2.4. Data Processing and Dynamic Emotion Feature Extraction

The data processing and analysis workflow in this study is illustrated in Figure 2A. First, based on the time segments in which participants performed the self-presentation task, video clips were extracted using FFmpeg (https://ffmpeg.org/download.html, accessed on 9 March 2025). FFmpeg is a powerful multimedia processing tool that has been widely applied in scientific research ([44]). Figure 2B presents a sample frame from the raw video data, showing the participant and the experimental environment. Next, we utilized the open-source tool OpenFace ([2]) to perform frame-by-frame facial recognition and annotate 68 facial landmarks, as shown in Figure 2C. Based on the detected landmarks, OpenFace further computed facial texture features to estimate AU scores. Detailed technical specifications are available on the official OpenFace documentation: https://github.com/TadasBaltrusaitis/OpenFace/wiki/Action-Units (accessed on 9 March 2025). Ultimately, OpenFace provided scores for 17 AUs (1, 2, 4, 5, 6, 7, 9, 10, 12, 14, 15, 17, 20, 23, 25, 26, and 45), each ranging from 0 to 5, with higher scores indicating greater AU intensity. A detailed description of the 17 AUs extracted using OpenFace is provided in the Appendix A. Figure 2D provides a time-series visualization of four selected AU scores, where the *x*-axis represents the frame number (time) and the *y*-axis represents AU intensity scores.

Building on previous research that has established the relationship between AUs and basic emotions ([27]; [48]; [57]), we computed frame-by-frame scores for each basic emotion by summing the corresponding AUs intensities. Generally, the self-presentation task is only weakly associated with two basic emotions: anger and surprise. Theorists suggest that anger typically arises in response to aversive conditions or undesired events ([37]), while surprise is triggered by unexpected occurrences ([28]). Since the self-presentation task does not involve stimuli that would plausibly induce these emotions, we focus only on the four basic emotions most relevant to self-presentation: happiness, sadness, fear, and disgust. In addition, we considered the 17 AUs identified by OpenFace alongside the emotion-to-AU mappings presented in Table 1. Based on this integration, we selected the AUs corresponding to the four basic emotions for emotional computation, as detailed in Table 2.

Finally, time-series analysis was conducted to extract dynamic features from the four basic emotional expressions. We implemented this analysis using the open-source TSFresh package ([7]) in Python 3.10 (https://www.anaconda.com/download, accessed on 9 March 2025), extracting 24 features for each emotion. These features encompass multiple dimensions, including statistical properties, dynamical characteristics, frequency-domain attributes, and linear trends, to comprehensively capture the temporal dynamics of emotional expressions. Statistical features quantify emotion intensity and distribution patterns. Dynamical features describe fluctuations and complexity in emotional expressions. Autoregressive and correlation-based features reveal temporal dependencies in emotional dynamics. Frequency-domain features capture oscillatory patterns and periodicity in emotional fluctuations. Linear trend features indicate the overall trajectory of emotional changes over time. A detailed description of the extracted features—including their names, psychological interpretations, computational methods, and parameter settings—can be found in Appendix A.

### 2.5. Machine Learning and Explainable AI

To model the complex and high-dimensional relationship between dynamic emotional features and self-esteem, we employed machine learning and explainable AI techniques to identify which dynamic features of momentary emotions are closely associated with self-esteem. Given the potentially intricate mapping between self-esteem and dynamic emotions, and the typically large sample sizes required for regression tasks, we opted for a classification approach, which generally requires a smaller sample size. It is important to note that our primary objective was to establish the relationship between dynamic emotional features and self-esteem, rather than to optimize model performance or compare different machine learning models. Therefore, we selected a single, well-established statistical learning model, the Support Vector Machine (SVM) with kernel methods ([8]). Kernel methods allow the SVM to effectively handle various data structures, ensuring the model’s robustness ([21]; [40]). In this study, we evaluated three different kernel functions to optimize the feature space representation and achieve the best performance: linear kernel, radial basis function (RBF) kernel, and polynomial kernel. In this study, the SVM with an RBF kernel and a regularization parameter of C = 1 demonstrated the best overall performance. The generalization results of SVMs with all three kernel functions are reported in Appendix A.

To ensure compatibility with the classification task, we discretized the continuous self-esteem scores. While the median split method is widely used in psychology and affective computing ([34]; [51]), it may obscure meaningful differences between high and low self-esteem groups ([26]). Therefore, to better highlight individual differences between groups, we adopted a top- and bottom-28% grouping strategy, resulting in a subset of 118 participants included in the machine learning classification task. In this study, participants with RSES scores below 28 were classified as low self-esteem (positive class, *N* = 59), while those with scores above 33 were classified as high self-esteem (negative class, *N* = 59). To evaluate model generalization more effectively and comprehensively, we report both classification accuracy (ACC) and F1-score as performance metrics. Additionally, we employed random 10-fold cross-validation to assess model generalizability, allowing us to determine whether dynamic emotional features are closely linked to self-esteem.

First, we built an SVM model using the RBF kernel that incorporated the dynamic features of all four basic emotions (happiness, sadness, fear, and disgust) to predict self-esteem classification labels. We then conducted model interpretability analysis using SHAP, which evaluates the marginal contribution of each feature to the model’s predictions. SHAP is a game-theoretic approach that quantifies both global and local feature importance by calculating the marginal impact of each feature across all possible feature combinations ([25]). In this study, we used the KernelExplainer method within SHAP to analyze 24 test samples (based on an 80/20 train-test split from 118 participants). For each participant, we extracted SHAP values for all 96 features (shape = [24, 96]). We then summed the absolute SHAP values across all samples for each feature and grouped the results according to the four emotion categories, thus obtaining the overall contribution of each emotion dimension to model predictions. Finally, we conducted ANOVA on the features with non-zero SHAP values to examine whether the contribution of different emotional categories differed significantly.

## 3. Results

Statistical analyses were conducted using SPSS 27. First, we conducted a correlational analysis on the full sample of 211 participants to examine the relationship between static emotions and self-esteem. Next, we performed a paired-samples *t*-test on the self-reported social evaluation threat scores of the 118 participants selected based on the top and bottom 28% of self-esteem scores, comparing pre- and post-task scores to assess the effectiveness of the public self-presentation manipulation. Building on this, we applied machine learning and explainable AI, aiming to investigate the relationship between dynamic features of different basic emotions and self-esteem. For the machine learning results, we first conducted a one-sample *t*-test on the model’s performance to determine whether it significantly exceeded the 50% random guessing level. This step was essential to verify whether the model effectively captured self-esteem. Finally, we conducted an ANOVA on the SHAP values to examine the relative contribution of the four basic emotion feature sets to the model’s predictions.

### 3.1. Static Emotional Representation of Self-Esteem: Correlation Analysis

The correlation analysis between self-esteem and long-term static emotions (PANAS scores) is summarized in Table 3. The results indicate that every PANAS item was significantly correlated with self-esteem. The overall pattern suggests that PA items were positively correlated with self-esteem, while NA items were negatively correlated with self-esteem.

### 3.2. Dynamic Emotional Representation of Self-Esteem: Explainable AI

A paired-samples *t*-test showed that participants’ post-task social evaluation threat scores (*M* = 4.38, *SD* = 20.3) were significantly higher than their pre-task scores (*M* = 3.76, *SD* = 1.76), *t*_(117)_ = 3.53, *p* < 0.001. This result indicates that the public self-presentation task effectively increased participants’ perceived social evaluation threat, reinforcing the role of self-esteem in this experimental context.

To assess the effectiveness of dynamic emotional features in representing self-esteem, we first established a model using the dynamic features of all four basic emotions (happiness, sadness, fear, and disgust). The model achieved a mean accuracy of 61.88% (±2.15%) and a mean F1-score of 63.95% (±2.56%). We then conducted a one-sample *t*-test comparing the classification accuracy against the random guessing level (0.5). The results showed that the model’s performance was significantly higher than random guessing, with ACC: *t*_(39)_ = 34.96, *p* < 0.001 and F1-score: *t*_(39)_ = 34.51, *p* < 0.001. This finding confirms that the dynamic features of these four basic emotions provide a valid and reliable representation of self-esteem.

The SHAP visualization results are presented in Figure 3. Among features that contributed meaningfully to the model (i.e., non-zero SHAP values), we conducted an ANOVA comparing the mean SHAP values of the four emotion categories (see Table 4 for descriptive statistics). The results showed no statistically significant main effect of emotional category, *F*_(3, 50)_ = 1.31, *p* = 0.281, $\eta_{p}^{2}$ = 0.073. This suggests that each emotional category contributed comparably to the model’s performance, indicating that the dynamic facial representation of self-esteem does not rely on a single type of emotion, but rather reflects a composite of multiple emotional signals.

## 4. Discussion

This study examined the relationship between self-esteem, static emotions, and dynamic emotional expressions. We recruited 211 participants and assessed self-esteem and long-term static emotions using the RSES and the PANAS, respectively. In addition, participants’ momentary emotional expressions during a public self-presentation task were captured using OpenFace. These expressions were then analyzed through multidimensional time-series analysis with TSFresh, allowing for the extraction of dynamic emotional features. Finally, machine learning and SHAP were conducted to achieve our research objectives.

### 4.1. Core Findings

Low self-esteem has long been recognized as an important risk factor for mental disease ([32]; [30]; [43]). Previous studies have established a strong link between self-esteem and psychopathology ([56]), suggesting that: (1) High self-esteem serves as a psychological resource, helping individuals buffer against the negative impact of life stressors; (2) Low self-esteem individuals are more susceptible to negative experiences, leading to greater long-term exposure to negative emotions and fewer positive emotional experiences. Our findings align with these perspectives and replicate the results of [6] ([6]), confirming that self-esteem is strongly associated with long-term stable emotions.

Moreover, our findings show that self-esteem is not only associated with long-term stable emotional states, but also closely linked to the dynamic facial expressions of four basic emotions (happiness, sadness, fear, and disgust). In contrast to previous research that primarily focused on emotional states derived from static self-reports (e.g., PANAS scores), our study used time-series analysis to capture the behavioral dynamics of facial expressions during a self-presentation task. This suggests that self-esteem is embedded not only in an individual’s long-term evaluation of self-worth, but also in their everyday emotional expression style.

From a theoretical standpoint, the Trait Activation Theory (TAT) offers a compelling explanation for this phenomenon. The TAT posits that personality traits must be activated by relevant situational cues in order to manifest as observable behavior ([46]). Terror Management Theory (TMT) provides a more detailed account of self-esteem’s function and specifies the contexts in which it is activated ([19]). According to TMT, self-esteem acts as a buffer against external threats, helping individuals mitigate anxiety and other negative emotions, especially in situations that threaten their self-worth. In this study, we introduced a public self-presentation task to elicit the social evaluation threat, aligning with TMT’s conceptualization of self-worth threats and thereby successfully activating self-esteem. In such a context, differences in emotional expression styles between individuals with high and low self-esteem became more pronounced. Moreover, the dynamic features of emotions—such as variability, trend, intensity, and frequency—are considered elements of one’s stable affective style, which has long been viewed as a behavioral manifestation of personality ([9]). Thus, subtle fluctuations in dynamic facial expressions not only reflect an individual’s current emotional state, but also reveal deeper psychological trait structures. Therefore, both positive and negative emotional expressions show significant associations with self-esteem.

Furthermore, the ANOVA results based on SHAP values indicate that the four basic emotions did not differ significantly in their contributions to predicting self-esteem—no single emotion outperformed the others. This finding suggests that the emotional representation of self-esteem is not driven by a dominant emotional category, but rather reflects a complex and multidimensional structure of emotional regulation, characterized by joint differences across multiple basic emotional dimensions. This expands on the traditional view linking self-esteem primarily to a single positive emotion (e.g., smiling or happiness) and highlights the need to understand the expressive features of self-esteem within a broader emotional system.

This multi-channel emotional differentiation pattern can be interpreted through the lens of Impression Management Theory (IMT; [24]) and self-presentation styles. According to IMT, individuals often strategically regulate their outward behavior (e.g., facial expressions, tone of voice) during social interactions to construct an idealized public image. Related studies have also shown that self-esteem is closely linked to self-presentation strategies: high self-esteem individuals tend to adopt self-enhancing styles, whereas low self-esteem individuals are more likely to engage in self-protective strategies, aiming to mitigate the psychological costs of evaluative failure ([33]). In the highly social context of our public self-introduction task, low self-esteem individuals may be more motivated to engage in impression management behaviors in advance, such as exaggerating smiles or suppressing negative emotions, to construct a socially desirable image of being “well-adjusted” and buffer against potential negative evaluations ([17]). Moreover, prior research highlights fundamental differences in the controllability of facial expressions. Macro-expressions typically last longer, are more intense, and are subject to conscious regulation, whereas micro-expressions are brief, automatic, and difficult to control voluntarily ([57]). By extracting dynamic time-series features of facial emotion, this study captured both components, enabling a more authentic depiction of emotional responses. More importantly, the association between self-esteem and a range of self-relevant negative emotional experiences has been well documented. Emotions such as shame, guilt, embarrassment, and fear are considered core elements of low self-esteem experiences ([45]; [49]). The facial expressions of these emotions not only convey an individual’s internal self-evaluation but also reflect their longing for social acceptance and sensitivity to rejection.

Therefore, in self-presentation contexts, individuals are likely to regulate multiple emotional expressions simultaneously, rather than simply increasing positive affect or suppressing negative emotions. These complex regulation patterns can be detected through dynamic, multidimensional facial expression data, and collectively contribute to the behavioral representation of self-esteem, as captured by machine learning models. These findings offer new insights into the behavioral representation of self-esteem.

### 4.2. Strengths, Limitations, and Future Directions

This study provides novel insights into the dynamic emotional representation of self-esteem, demonstrating that self-esteem not only influences long-term emotions but also manifests in spontaneous, momentary dynamic emotions. Previous research has primarily relied on static emotion assessment methods (e.g., self-reported PANAS), which depend on introspective evaluation ([6]). By utilizing dynamic facial expression features, our study overcomes this limitation, extending self-esteem research from static emotional representations to natural, dynamic emotional expressions. This advancement offers a new perspective on the relationship between self-esteem and emotional expression. Moreover, the study innovatively employs dynamic facial expression features to represent self-esteem, breaking away from traditional self-esteem assessment methods (e.g., self-report scales) and offering a new approach to self-esteem evaluation based on nonverbal behavior. Compared to traditional scale-based measurements, our method is more objective, reducing self-report biases and providing a nonverbal assessment of self-esteem.

Despite these theoretical and methodological contributions, this study has several limitations that should be addressed in future research. First, our experimental paradigm constrained the range of emotional expressions. While the public self-presentation task is a standard paradigm for studying self-esteem ([20]), it did not elicit emotions such as surprise and anger. However, anger has also been shown to be closely related to self-esteem ([18]; [23]). Future studies could explore alternative paradigms, such as more naturalistic social interactions or high-ecological-validity social feedback scenarios, to capture a broader spectrum of emotional responses.

Second, participants in this study engaged in self-presentation in front of strangers, whereas in real-life social interactions, emotional expressions may differ when interacting with familiar individuals, such as friends or family. For example, low self-esteem individuals may not exhibit as much fear in familiar social contexts. Future research could compare the relationship between self-esteem and dynamic emotional expressions across different social contexts, enhancing the generalizability of the findings.

Third, this study used OpenFace to extract AUs to compute basic emotion scores, rather than employing end-to-end deep learning models (e.g., DeepFace). This decision was based on two key considerations: (1) Frequent mouth movements during the self-presentation task (e.g., speaking, opening the mouth) could interfere with end-to-end models to recognize facial expression, potentially reducing emotion recognition accuracy. (2) End-to-end models produce normalized probability distributions as output (i.e., the sum of all emotion scores equals 1), rather than absolute emotion intensity values. This normalization process complicates time-series analysis, as it distorts the dynamic patterns of individual emotions over time. By computing emotion scores from AUs, our approach reduces the impact of mouth movements, enhances emotion recognition stability, and preserves the continuity of emotional signals, making it more suitable for time-series analysis. It is important to note, however, that the relationship between AUs and emotional expressions is not one-to-one. There is inherent ambiguity and overlap, whereby a single emotion may correspond to multiple AUs, and one AU may contribute to multiple emotions ([27]; [57]). The mapping between AUs and emotions remains debated within the academic community. In this study, we adopted a simplified AU summation strategy to reduce methodological complexity. We acknowledge that this approach introduces measurement error in the representation of emotions. Future research could improve AUs-to-emotion mapping methods to enhance emotion recognition accuracy. Alternatively, advanced end-to-end emotion recognition models could be developed to enable real-time emotion detection without interference from mouth movements.

Fourth, although our sample size of 211 participants is considered moderate in psychological research, it remains relatively small for complex personality recognition tasks. Particularly in the machine learning classification task, the dataset was further reduced to 118 samples after applying the high/low self-esteem grouping strategy. This relatively small sample size limits the overall model performance—despite statistical significance, ACC reached only 61.88% and an F1-score of 63.95%, reflecting the model’s constrained predictive power. Additionally, the gender distribution was uneven (33 males, 178 females), which may limit the generalizability of our findings across genders. Future research should aim to increase the sample size and balance gender proportions to enhance the robustness and applicability of the results.

Finally, given the cultural sensitivity of self-esteem and the multimodal complexity of its expression, future research should adopt cultural and multimodal perspectives to comprehensively investigate the relationship between self-esteem and dynamic emotional expression.

## 5. Conclusions

This study reveals the relationship between self-esteem and the dynamic features of momentary emotional expressions. Our findings demonstrate a robust association between self-esteem and the expression of four basic emotions (happiness, fear, sadness, and disgust) without any single emotion showing a statistically dominant contribution. This suggests that self-esteem may not rely on the expression or suppression of a specific emotion but instead reflects a complex and multidimensional emotional regulation pattern. This finding expands on the traditional perspective that high self-esteem is primarily associated with the expression of positive emotions, such as increased smiling. Instead, our results suggest that both positive and negative emotional expressions jointly contribute to the behavioral manifestation of self-esteem. Therefore, the study introduces a new perspective on self-esteem assessment, highlighting the potential of nonverbal behavioral indicators as alternatives to traditional self-report measures.

## Figures and Tables

**Figure 1 behavsci-15-00709-f001:**
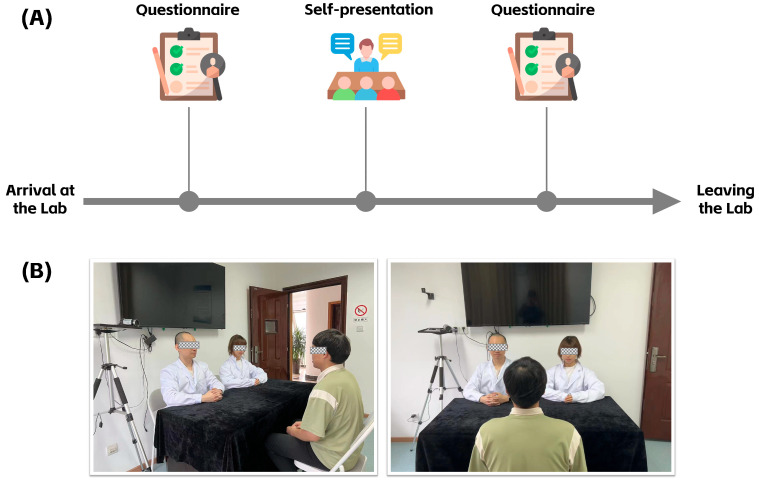
(**A**) Experimental procedure flowchart. Upon arrival at the laboratory, participants first completed a series of questionnaires. They then performed the public self-presentation task. After completing the task, participants filled out a second set of questionnaires before leaving the laboratory. (**B**) Laboratory environment and setup. The researchers, one male and one female, sat directly opposite the participant. Participants were required to introduce themselves to the two researchers, and the entire process was recorded by the camera.

**Figure 2 behavsci-15-00709-f002:**
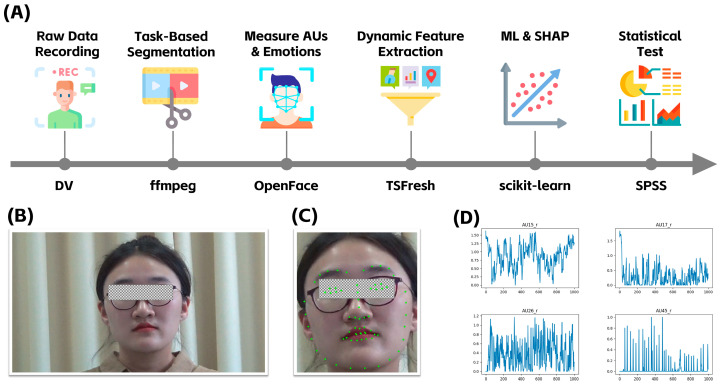
(**A**) Flowchart of data processing and analysis. First, video recordings of the experimental task were segmented, and Facial Action Units (AUs) were annotated using OpenFace. Next, instantaneous emotion scores were computed based on the AUs, and dynamic features were extracted using TSFresh. Finally, machine learning (ML) and SHapley Additive exPlanations (SHAP) were conducted to examine the relationship between dynamic emotions and self-esteem. (**B**) Example of raw video footage showing the participant and the experimental environment. (**C**) Visualization of facial landmarks annotated by OpenFace. (**D**) Time-series visualization of four selected AUs. The *x*-axis represents the frame number (time), while the *y*-axis represents the intensity scores of the respective AUs.

**Figure 3 behavsci-15-00709-f003:**
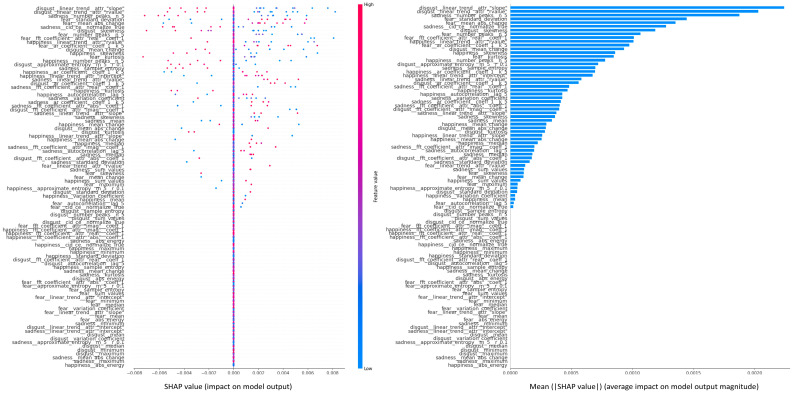
SHAP-based global feature interpretation. This figure presents the feature importance results for all 96 dynamic emotional features in the SVM model, generated using SHAP. The left panel shows a SHAP beeswarm plot, where the *x*-axis represents the marginal impact of each feature on the prediction of self-esteem, and the *y*-axis lists all features; color indicates the original feature value. The right panel displays a bar plot of the average absolute SHAP value for each feature, indicating its overall importance across all samples.

**Table 1 behavsci-15-00709-t001:** Description of facial muscles and other nonverbal behaviors involved in the emotions.

Emotion	Darwin’s Description (Nonfacial Elements in Parentheses)	AUs Found to Be Associated with This Emotion in Research with Humans (Optional AUs in Parentheses)
Anger	Nostrils raised, mouth compressed, furrowed brow, eyes wide open, head erect (chest expanded, arms rigid by sides, stamping ground, body swaying backward/forward, trembling)	4; 5 or 7; 22; 23; 24
Disgust	Lower lip turned down, upper lip raised, expiration, mouth open, spitting, blowing out, protruding lips, throat-clearing sound, lower lip and tongue protruding	9 or 10; (25 or 26)
Fear	Eyes open, mouth open, lips retracted, eyebrows raised (crouching, paleness, perspiration, hair standing on end, muscles shivering, yawning, trembling)	1; 2; 4; 5; 20; (25 or 26)
Happiness	Eyes sparkling, skin under eyes wrinkled, mouth drawn back at corners	6; 12
Sadness	Corners of mouth depressed inner corner eyebrows raised (low spirits)	1; (4); 15; (17)
Surprise	Eyebrows raised, mouth open, eyes open, lips protruding (expiration, blowing/hissing, open hands high above head, palms toward person with straightened fingers, arms backwards)	1; 2; 5; 25 or 26

**Table 2 behavsci-15-00709-t002:** Computation of basic emotions from AUs and their psychological interpretation in the self-presentation task.

Emotion	Psychological Interpretation in the Self-Presentation Task	AUs-Based Computation of Instantaneous Emotion Score
Happiness	Experienced when participants mention their strengths or things they enjoy.	AU06 + AU12
Sadness	Experienced when participants talk about their weaknesses or sad experiences.	AU01 + AU15
Disgust	Experienced when participants discuss things they find aversive or unpleasant.	AU09 + AU10
Fear	Induced by social evaluation threat during public self-presentation.	AU01 + AU04 + AU05 + AU20

**Table 3 behavsci-15-00709-t003:** Pearson’s correlation coefficients between self-esteem and PANAS items.

Items	*r*	Items	*r*
Interested	0.406 ***	Irritable	−0.224 ***
Distressed	−0.484 ***	Alert	0.304 ***
Excited	0.385 ***	Ashamed	−0.333 ***
Upset	−0.374 ***	Inspired	0.286 ***
Strong	0.414 ***	Nervous	−0.431 ***
Guilty	−0.386 ***	Determined	0.451 ***
Scared	−0.336 ***	Attentive	0.437 ***
Hostile	−0.170 *	Jittery	−0.296 ***
Active	0.430 ***	Enthusiastic	0.463 ***
Proud	0.539 ***	Afraid	−0.454 ***

Note: * *p* < 0.05, *** *p* < 0.001.

**Table 4 behavsci-15-00709-t004:** Mean SHAP values and number of non-zero features for the four basic emotions.

Basic Emotions	|SHAP Value|		Number of Non-Zero Features
	*M*	*SEM*	
Disgust	0.019	0.004	11
Fear	0.014	0.004	12
Happiness	0.010	0.003	15
Sadness	0.012	0.003	16

## Data Availability

The data and code presented in this study are openly available in GitHub at https://github.com/Terryzang/time-series-analysis-AUs (accessed on 9 March 2025).

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
