# Peer review of "Dynamic Facial Emotional Expressions in Self-Presentation Predicted Self-Esteem"

_behavsci, 2025, doi:10.3390/bs15050709_

Round 1

Reviewer 1 Report

Comments and Suggestions for Authors

Dear Authors, 

I read the manuscript titled "Dynamic Facial Emotional Expressions in Self-Presentation predicted self esteem with interest. I commend your efforts in developing such a rich dataset, which can be used to investigate many interesting questions about the interactions between personality and emotional expressions during social interactions. While the dataset and the preprocessing efforts are impressive, I was not completely convinced about the results described in the manuscript.  I had the following three concerns about the analysis:

1. Self-esteem is an inherently continuous variable. Performing a median split discards fine-grained differences in self-esteem. It bundles individuals who are close to and far away from the threshold, say with a score of 1 vs. 30, into the same group. Similarly, it bundles someone with a score of 30 and someone with a score of 31 into different groups, whereas they are actually quite similar phenotypically. Performing a median split is valid if the distribution of self-esteem scores is bimodal or if there is extensive literature suggesting that the population distribution of self-esteem demonstrates different phenotypes based on whether their score is >= 30. As a result of this arbitrary classification, the SVM also seems to have a really poor categorization accuracy (the baseline model is 55% accurate, where chance is 50%), which leads to my next point. Please also refer to the following reference for a more thorough discussion of this issue: MacCallum, R. C., Zhang, S., Preacher, K. J., & Rucker, D. D. (2002). On the practice of dichotomization of quantitative variables. Psychological Methods, 7(1), 19–40

2 . The validity of ablation experiments depends on the accuracy of the original model. For some reason, the models presented here did not really learn to distinguish the two classes well. One possibility for this is that the two groups do not sufficiently differ in their mean self-esteem scores. Another reason could be that the kernel used for SVM may not be appropriate, which brings me to the third and most crucial issue. 

3. Ablation experiments comparing the removal of one feature to the baseline model have a clear interpretation when the SVM uses a linear kernel. When the SVM employs a polynomial kernel or a more complex RBF kernel, the model can also capture non-linear interactions between features, which compromises the clean interpretation of the role of each feature on the loss of model performance. This can be remedied using techniques such as Permutation Importance or SHAP. However, these methods will still not mitigate issues arising from the poor overall performance of the model. 

Given these critical issues, I cannot recommend this manuscript for publication in the current form. 

Reviewer 2 Report

Comments and Suggestions for Authors

This study revealed the relationship between self-esteem and the dynamic features of momentary emotional expressions.. The paper is well written, describing in a good manner a problematic situation. The paper fits on the journal scope. There are some aspects could be improved:

  • Include related work section. Check the work of Mendez et al, (In Spanish),  Rueda de emociones de Ginebra+: instrumento para la valoración emocional de los usuarios mientras participan en una evaluación de sistemas interactivos
  • Analyze evaluation using multimodal tecniques
  • Analyze aspects related with culture
  • How to include AI on the process

Reviewer 3 Report

Comments and Suggestions for Authors
  1. (2.3. Experimental Measures) Adding a workflow diagram illustrating the full procedure of the Experimental Measures would enhance the clarity and specificity of the research methodology.
  2. (2.4.) 
  • Provide a detailed explanation of how AU scores are calculated by OpenFace and how they were applied in the present study.
  • Clarify the number of AUs used in the analysis and the rationale for their selection.
  • Include analysis and interpretation based on the selected AUs, such as how they contribute to emotion-specific predictions.
  • For each target emotion, explain the role that the selected AUs play in its expression, referencing relevant psychological or FACS literature.
  • Clearly state whether emotional scores were derived through simple summation of AU values or using a more structured method, and justify the chosen approach.

3. Results

  • Please consider presenting the classification performance of each SVM kernel (e.g., linear, RBF, polynomial) used in the comparative analysis. This would provide clearer evidence for the selection of the polynomial kernel and enhance the transparency of the model evaluation process.
  • Additionally, there appears to be a typographical duplication in the heading or reference to Table 3.

4. Discussion

  • While the study provides a meaningful analysis of how dynamic emotional features contribute to self-efficacy prediction and highlights the relative importance of emotions such as sadness, fear, and disgust through ablation experiments, the overall classification performance remains modest. Although the baseline model shows statistically significant results (accuracy = 55.92%, p < 0.001), the practical utility of using emotional dynamics alone appears limited. Further interpretation and discussion of this limitation would strengthen the manuscript.

Round 2

Reviewer 1 Report

Comments and Suggestions for Authors

I commend the authors for making the analysis more transparent and rigorous and making the discussion more nuanced. Due to their efforts, I now have a clearer picture of the results. The results indicated that all four basic emotions are closely associated with self-esteem, with disgust showing the strongest association. However, I still feel that the inclusion of the ablation experiment (and not the SHAP analyses in the main manuscript) does not amount to evidence that can support the claims made in the discussion. In their response, the authors explain that including feature-wise SHAP analyses can detract from the paper's central theme. However, I believe the SHAP analyses are critical for the authors' claim that "... the regulation of negative emotional expressions, especially disgust, plays a central role in reflecting self-esteem, potentially more so than the display of positive emotions" (lines 412-415). Crucially, the ablation analyses, which have been expanded and included in the revised manuscript, cannot directly support this claim. 

The technical reason for this is that when using a non-linear kernel such as the RBF for SVM, ablation experiments showing a reduction in performance when removing one feature are not guaranteed to imply that that specific feature is an important predictor. It could also be an interaction of that feature with the other features. Interpretable machine learning techniques such as SHAP exist precisely for this purpose - to show the direct contribution of each feature while accounting for feature interactions. Thus, if the central theme of the paper is to show the importance of negative self-emotions vs. positive self-emotions, the authors can show an additional plot by lumping together the SHAP values of the negative features and showing that the average SHAP score of the negative features is higher than that of the positive self-emotion feature scores. A quick look at the SHAP score charts suggests this may work, but I cannot be sure without knowing the statistical test results. Without such analysis, the connection between the computational findings and the theoretical interpretations made in the discussion remains tenuous. 

To summarize, I believe that by performing sampling from the tails rather than a median split and applying an RBF kernel, the authors have successfully addressed concerns (1) and (2) in my original review; however, I remain unconvinced by their response to concern (3). My recommendation is to include the complete SHAP analysis and an additional planned comparison between average negative self-emotion SHAP scores and positive self-emotion SHAP scores in the main manuscript. 

Round 3

Reviewer 1 Report

Comments and Suggestions for Authors

I am satisfied with the new version of the manuscript and am happy to recommend acceptance.